# Multiparameter Flow Cytometric Analysis of the Conventional and Monocyte-Derived DC Compartment in the Murine Spleen

**DOI:** 10.3390/vaccines12111294

**Published:** 2024-11-19

**Authors:** Ronald A. Backer, Hans Christian Probst, Björn E. Clausen

**Affiliations:** 1Institute for Molecular Medicine, Paul Klein Center for Immune Intervention, University Medical Center of the Johannes Gutenberg-University Mainz, 55131 Mainz, Germany; 2Research Center for Immunotherapy (FZI), University Medical Center of the Johannes Gutenberg-University Mainz, 55131 Mainz, Germany; 3Institute for Immunology, Paul Klein Center for Immune Intervention, University Medical Center of the Johannes Gutenberg-University Mainz, 55131 Mainz, Germany

**Keywords:** dendritic cell, dendritic cell subpopulations, macrophages, monocytes, multicolor flow cytometry, myeloid cells, spleen

## Abstract

Dendritic cells (DCs) are present in almost all tissues, where they act as sentinels involved in innate recognition and the initiation of adaptive immune responses. The DC family consists of several cell lineages that are heterogenous in their development, phenotype, and function. Within these DC lineages, further subdivisions exist, resulting in smaller, less characterized subpopulations, each with its unique immunomodulatory capabilities. Given the interest in utilizing DC for experimental studies and for vaccination purposes, it becomes increasingly crucial to thoroughly classify and characterize these diverse DC subpopulations. This understanding is vital for comprehending their relative contribution to the initiation, regulation, and propagation of immune responses. To facilitate such investigation, we here provide an easy and ready-to-use multicolor flow cytometry staining panel for the analysis of conventional DC, plasmacytoid DC, and monocyte-derived DC populations isolated from mouse spleens. This adaptable panel can be easily customized for the analysis of other tissue-specific DC populations, providing a valuable tool for DC research.

## 1. Introduction

The dendritic cell (DC) family comprises a remarkably diverse group of cells that, despite distinct developmental characteristics, share critical functions such as innate pathogen detection, antigen (Ag) presentation, and cytokine production. As such, DCs play pivotal roles in orchestrating a broad spectrum of innate and adaptive immune responses towards pathogens, allergens, and cancer. Broadly categorized, DCs fall into three distinct classes: conventional (or classical) DCs (cDCs), plasmacytoid DCs (pDCs), and monocyte-derived DCs (MoDCs). Among these, cDCs are the most potent Ag-presenting cells, playing crucial roles in eliciting Ag-specific CD4^+^ and CD8^+^ T cell responses, as well as the initiation of innate immunity [1,2,3,4,5]. On the other hand, while pDCs are poor in Ag presentation, they exhibit specialization in type-I interferon production, particularly during viral infection [6,7]. Meanwhile, MoDCs predominantly emerge in specific inflammatory conditions, serving as an emergency source of DC at the site of infection [8,9].

Current DC research focuses on harnessing individual DC populations for cancer immunotherapy and vaccination strategies [10]. However, the phenotypic overlap among several DC populations complicates the identification of their specific immunomodulatory functions. This is particularly evident with type 2 cDC (cDC2) and MoDC, whose phenotypes often converge, especially during inflammation, causing confusion regarding their individual immune regulatory functions [8,11,12,13,14,15]. Additionally, our understanding of the phenotypic and functional heterogeneity within the cDC1/cDC2 populations, including the existence of tissue-specific subsets and their precise functions, remains limited [1,12,16,17,18]. Therefore, the discrimination of the DC network into clearly delineated lineages is crucial for gaining a more complete understanding of how individual DC subsets contribute to the regulation of immune responses. To this end, here, we present an easy and ready-to-use protocol for flow cytometric analysis of the distinct DC populations in the murine spleen, encompassing cDC1 and cDC2 subpopulations, pDC, and monocyte/macrophage populations.

**Objective and purpose of this protocol:** While recent technical advances, particularly single-cell RNA sequencing approaches, have facilitated the identification of novel DC lineage-specific markers, many of these markers, such as transcription factors, do not always enable the precise and satisfactory distinction of DC subsets by flow cytometry, especially during functional in vivo and in vitro studies. This protocol utilizes a conserved 26-color core-marker panel to achieve a precise and refined phenotypic characterization of these cells, thereby greatly advancing our understanding of the distinct DC lineages. Given that the DC network is highly conserved, our analysis may also serve as a robust basis for identifying DCs across various tissues with minimal modifications.

## 2. Materials and Methods

Our DC staining panel was optimized for use with the BD Biosciences FACSymphony™ A5 cell analyzer, featuring five lasers (355 nm, 405 nm, 488 nm, 561 nm, and 637 nm) and up to 50 detectors. The materials and methods, including the antibody details and dilutions used, are provided in Table 1 and Table 2, as well as the Appendix A.

### 2.1. Mice

Eight-week-old male C57BL/6 mice were housed at a barrier-free and specific pathogen-free facility at the Translational Animal Research Center of the University Medical Center Mainz (Mainz, Germany). All mice were used in accordance with institutional and national animal experimentation guidelines.

### 2.2. Preparation of Single-Cell Suspension from Spleens

DCs are typically present in low numbers within the tissue, where they reside as highly adherent cells that are tightly embedded within the extracellular tissue matrix. Consequently, enzymatic digestion of the tissue is a critical step in DC research, because DCs are very sensitive to mechanical stimuli and endotoxin contamination.

The isolation of splenic DCs by Collagenase Type IV and DNase-I yields a single-cell suspension with a high number of DCs (typically 3–5 × 10^6^ CD11c^+^ DCs per spleen), while preserving their immature phenotype and cell surface markers for consistent and reproducible flow cytometric characterization [19]. For this, mice are euthanized with isoflurane and spleens are collected. After removing any fat tissue, the spleens are cut into grain-sized pieces. These spleen pieces are then incubated in 1 mL RPMI medium containing 200 U/mL of Collagenase Type IV and 0.5 U/mL of DNaseI, with agitation at 37 °C for 30 min or until fully digested. Upon completion of the incubation period, EDTA is added to achieve a final concentration of 10 mM, and the cell suspension is further incubated for 5 min at 4 °C. Undigested material is filtered out, while larger fragments of tissue can be forced through a filter using a 1ml syringe plunger. Subsequently, red blood cells are lysed, and the remaining cells are washed once with PBS supplemented with 10 mM EDTA. If required, cellular debris can be removed by additional filtration steps.

### 2.3. Antibody Staining of Single-Cell Suspensions from Splenic Tissue for Flow Cytometry

#### 2.3.1. Cell Preparation, Antibody Mixtures, and Staining Procedure

Please note that detailed information for the staining kit, as well as used buffers and their reagents, are listed as Appendix A.

Calculate and prepare the first Antibody Master Mix in Brilliant Stain Buffer Plus (BD Biosciences, Franklin Lakes, NJ, USA; order number 566385) containing all antibodies for staining extracellular markers (excluding anti-Langerin antibody). Use 50 µL of Brilliant Stain Buffer Plus per sample and add the antibodies, as specified in Appendix A, at the indicated dilution. Be sure to mix the Master Mix regularly while pipetting the antibodies to avoid clumping.

Count the cells, centrifuge at 400× *g* at 4 °C, and resuspend at a concentration of 20 × 10^6^ cells/mL in FACS buffer. Plate 75 μL of the cell suspension (1.5 × 10^6^ cells) per well in a U-bottom 96-well plate, including controls and optional stainings for instrument setup.

Centrifuge the plate containing cells at 400× *g* for 5 min at 4 °C. Discard the supernatant and tap the plate upside down on a paper towel. Add Fc-block solution (BioXCell, Lebanon, NJ, USA; order number BE0008; 1:100 dilution) to the cells. Mix thoroughly by pipetting and incubate for 15 min at room temperature. Centrifuge the plate again at 400× *g* for 5 min at 4 °C, discard the supernatant, and tap the plate upside down on a paper towel. Next, apply 50 μL of the prepared first Antibody Master Mix onto the cells, mix by pipetting, and incubate for 30 min at 4 °C in the dark. Afterwards, fill the wells with 150 µL FACS buffer, centrifuge at 400× *g* for 5 min at 4 °C, discard the supernatant, and tap the plate upside down on a paper towel. Finally, wash the cells by adding 250 μL of FACS buffer before resuspending and centrifugating at 400× *g* for 5 min at 4 °C.

#### 2.3.2. Intracellular Langerin Staining

For intracellular antibody staining, fixate cells by adding 200 μL Fix/Perm buffer per well. Incubate for at least 30 min at room temperature in the dark. While incubating, prepare the intracellular antibody staining mix. After the fixation step, centrifuge the plate at 400× *g* for 5 min at 4 °C. Discard the supernatant and carefully tap the plate upside down on a paper towel. Wash the cells twice by adding 200 μL Perm/Wash buffer before resuspending and centrifuging again at 400× *g* for 5 min at 4 °C. Add 50 μL of the intracellular antibody mix to the cells, mix by pipetting, and incubate for 45 min at 4 °C in the dark. Fill the wells with 150 µL Perm/Wash buffer, centrifuge at 400× *g* for 5 min at 4 °C, discard the supernatant, and tap the plate. Finally, wash the cells by adding 250 μL Perm/Wash buffer before resuspending and centrifugation again, repeating the wash step.

#### 2.3.3. Sample Acquisition

Resuspend the samples in 200 µL FACS buffer. Store samples at 4 °C in the dark for use within the next 24 h. After performing the cytometer setup, filter the sample via 70 µm cell filters (Greiner bio-one, Austin, TX, USA; order number 542070) into FACS tubes. Acquire the data with an appropriate flow cytometer. For the current manuscript, a 5-laser BD FACSymphony^TM^ A5 instrument was used (BD Biosciences, Franklin Lakes, NJ, USA). Additional information on the used laser lines, mirrors, and optical filters can be found in the Appendix A. Data were collected after automated color compensation and analyzed using FlowJo software (BD Biosciences, version 10.10.).

## 3. Results

Our standardized 26-color core-marker FACS panel enables the precise identification of the major leukocyte and cDC populations in the steady-state murine spleen (Figure 1A). Specifically, this panel facilitates the classification of splenocytes into distinct lymphocyte populations (CD4^+^ and CD8^+^ T cells, B cells) as well as myeloid cell populations (DC, monocytes, macrophages). Additionally, it allows for the differentiation of the highly heterogeneous cDC population into multiple discrete but less characterized XCR1^+^ cDC1 and SIRPα^+^ cDC2 subpopulations (Figure 1B). In this staining panel, we used CD90.2 (Thy1.2) as the pan T cell marker because, unlike CD3, it is not downregulated on activated T cells. CD90.2 can be used for the most common inbred mouse strains, including C57BL/6, BALB/c, and CH3.

### 3.1. Characterization of Dendritic Cell Subsets in Steady-State Mouse Spleen

A representative gating strategy for the identification of multiple splenic DC subsets is shown in Figure 2. After excluding dead cells, using the Fixable Viability Dye FVS780, as well as doublets and debris, immune cells were identified using the pan-hematopoietic marker CD45. From this CD45^+^ cell population, T cells (CD90.2^+^), B cells (CD19^+^), and NK/NKT cells (NK1.1^+^CD49b^+^) were eliminated to enrich for DC, macrophages, and monocytes, collectively marked as lineage-negative cells (Figure 2A).

Within the lineage-negative cell population, macrophages are identified as F4/80^+^CD64^+^ cells (Figure 2B). Among the remaining cells, cDCs express MHC class II (MHC-II) and CD11c (Figure 2C). As macrophages can express these prototypical cDC markers as well, bona fide cDCs can be separated from contaminating macrophages by staining for CD26 and CD64 expression, respectively (Appendix A). Subsequently, pDCs are identified based on their PDCA1 expression, characterized further by low CD11c expression (Figure 2D), and can be further classified as MHC-II^lo^B220^hi^CD45RA^+^Siglec-H^+^ cells (Table 3). Based on the expression of CD8α and -β, pDCs can be further divided into three subsets [20,21]. The CD8α^+^ subset, involved in the induction of FOXP3^+^ regulatory T cells, is present in the spleen, lung, and lymph nodes (LNs).).

**Figure 2 vaccines-12-01294-f002:**
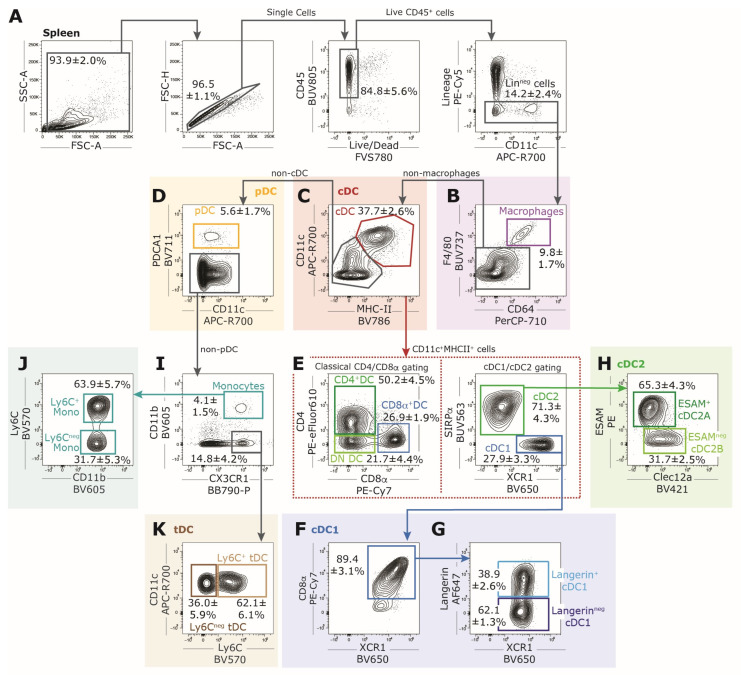
**Flow cytometry analysis of the steady-state splenic myeloid cell network.** Splenocytes from 8-week-old mice were stained with the described DC panel (Table 1 and Appendix A). (**A**) Cells were initially gated on single cells using FSC-A and SSC-A gating to eliminate doublets and to exclude debris. Subsequently, living CD45^+^ leukocytes were selected using viability-negative gating, followed by the exclusion of T cells (CD90.2^+^ cells), B cells (CD19^+^ cells), and NK/NKT cells (NK1.1^+^, CD49b^+^ cells). (**B**) Within this lineage-negative cell population, the expression of F4/80 and CD64 marked splenic red pulp macrophages. (**C**) Subsequently, cDCs are characterized as cells expressing MHC class II (MHC-II) and CD11c. (**D**) pDCs were identified as positive for PDCA1, while displaying intermediate levels of CD11c. Additionally, pDCs displayed variable expression of CD8α, which might indicate differences in the activation status of these cells. Splenic cDCs can either be separated into CD4^+^CD8α^−^, CD4^−^CD8α^+^ and double-negative DCs (**E, *left***), or, alternatively, into conventional cDC type 1 (cDC1; XCR1^+^; SIRPα^−^) and type 2 (cDC2; XCR1^−^; SIRPα^+^) subsets (**E, *right***). (**I**) Monocytes were identified as CD11c^−^CX3CR1^+^CD1b^+^ cells among the remaining cells and comprise Ly6C^+^ (‘pro-inflammatory’) and Ly6C^−^ (‘patrolling’) populations (**J**). (**F**) Terminally differentiated cDC1s expressed CD8α and (**G**) consisted of Langerin^+^ and Langerin^−^ subpopulations. (**H**) cDC2s comprised two dominant lineages, cDC2As and cDC2Bs, based on the mutually exclusive expression of ESAM and Clec12A, respectively. (**K**) Next to cDC and pDC the newly characterized non-canonical transitional DC (tDC) are identified as CD11b^−^CX3CR1^+^ cells. tDC are further divided into Ly6C positive (‘pDC-like’) and Ly6C negative (‘cDC-like’) populations, each with different functional characteristics. Depicted is one exemplary gating strategy. Data acquisition was performed using a BD FACSymphony flow cytometer, and data analysis was conducted using FlowJo software.

**Table 3 vaccines-12-01294-t003:** Summary of marker expression on analyzed cell populations.

Cell Population	Marker-Negative	Marker-Positive
cDC1	Lineage, CD11b, SIRPα, Ly6G, Siglec-F	CD8α, CD11c, CD24, CD26, CD205, MHC-II, XCR1, Dectin-1^+/−^
Langerin^+^ cDC1		CD103, Langerin
cDC2	Lineage, CD8α, CD24, CD205, XCR1, Ly6G, Siglec-F	CD11c, CD26, CD11b, MHC-II, SIRPα, Dectin-1^+/−^
cDC2A		CD4, ESAM
cDC2B		Clec12A, CX3CR1, Ly6C
Plasmacytoid DCs (pDCs)	Lineage, MHC-II, CD64	Ly6C, PDCA-1, SIRPα, CD8α^+/−^
Macrophages	Lineage, CD26, Ly6G, Siglec-F, XCR1	CD11c, CD11b, CD64, F4/80, FcεR1a, MHC-II, MerTK, SIRPα
Monocytes	Lineage, CD11c, MHC-II, Ly6G, Siglec-F	CD11b, CD64, CX3CR1, Ly6C^+/−^
Mo-DC	Lineage, CD26, Ly6G	CD11c, CD11b, CD64, Ly6C, MHC-II, SIRPα
Inflammatory DC (Inf-cDC2)	Lineage, XCR1	CD11c, CD26, CD64, FcεR1a, MHC-II, SIRPα

*Lineage: L/D, CD19, CD49b, CD90.2, NK1.1; +/−: subset- or activation-status-dependent expression*.

Historically, cDCs were, analogous to T cells, subdivided into different populations based on CD4 and CD8α surface expression (note that cDCs express CD8 as an αα homodimer and not, like T cells, as an αβ heterodimer) (Figure 2E, *left*). Following this classification, CD4^+^CD8α^−^ and double-negative cDCs both express CD11b and were often termed myeloid-related DCs. In contrast, CD4^−^CD8α^+^ cDCs are CD11b-negative and were referred to as lymphoid-related cDCs [22]. To date, cDCs are rather categorized into type 1 (cDC1) and type 2 (cDC2) populations based on the expression of XCR1 and SIRPα, respectively [11,23] (Figure 2E, *right*). Here, XCR1^+^ cDC1s correspond to the CD4^−^CD8α^+^ cDC subset, while SIRPα^+^ cDC2s are more heterogeneous and include both CD4^+^CD8α^−^ and CD4^−^CD8α^−^ cDCs. Generally, cDC1s are crucial for type 1 immune responses during viral infection and tumor surveillance [24]. Although the exact function of cDC2s is less clear and likely tissue-dependent, they are implicated in priming type 2 CD4^+^ T cell responses [4,25,26] and type 3 immunity by activating ILC3s and Th17 cells [4,13,27].

***XCR1^+^ cDC1s:*** In humans, both a population of XCR1^+^ and XCR1^−^ cDC1s can be identified [28]. Here, XCR1 expression marks terminally differentiated cDC1s that display a more mature, pre-activated phenotype and are the predominant subset secreting pro-inflammatory cytokines upon TLR stimulation [28]. Although XCR1 expression on murine cDC1s is rather homogenous and XCR1^+^ cDC1s represent a population that makes up approximately 30% of all splenic cDCs (Figure 2E). These cells express CD8α, while the small fraction of CD8α^−^ cells within this population characterizes less differentiated precursors (Figure 2F). Additionally, in the spleen, a distinct Langerin/CD207^+^ cDC1 subset exists (Figure 2G). While these Langerin^+^ cDC1s are considered a final cDC1 maturation stage, their ontogeny and subset relationship lack definitive evidence. Functionally, Langerin^+^ cDC1s play crucial roles in the cross-presentation of circulating apoptotic cells, IL-12p70 production, and the clearance of viral and bacterial pathogens, whereas the functions of Langerin-negative cDC1s are less well-defined [29,30,31,32,33]. Importantly, cell surface expression of Langerin differs on splenic cDC1s among inbred mouse strains. While Langerin is detectable on splenic cDC1s from BALB/c mice, its expression is nearly absent on splenic cDC1s from C57BL/6 mice, as Langerin expression on these cells occurs primarily intracellularly [34]. Apart from the shared expression of CD24, CD26, and CD205, Langerin^+^ cDC1s can also be distinguished from their Langerin-negative counterparts by the surface expression of CD103 (Figure 3A).

***SIRP**α^+^ cDC2s:*** cDC2s comprise a heterogeneous population, with CD11b^+^SIRPα^+^ cDC2s representing about 70% of splenic cDCs (Figure 2E). Recent studies suggest that cDC2s consist of at least two predominant lineages characterized by mutually exclusive expression of the transcription factors T-bet and RORγt [12]. T-bet^+^ cDC2As are distinguished by high ESAM expression, while RORγt^+^ cDC2Bs are ESAM^−^ and selectively express Clec12A (Figure 2H). ESAM^+^ cDC2As are proficient in MHC-II Ag presentation and CD4^+^ T cell activation but exhibit limited production of pro-inflammatory cytokines, a trait more pronounced in ESAM^−^ cDC2Bs [13,35,36]. ESAM^−^ cDC2s preferentially express monocyte-associated markers, such as CX3CR1 and Ly6C, and lower levels of the cDC marker CD26 than ESAM^+^ cDC2s (Figure 3B). As a consequence, ESAM^−^ cDC2s have sometimes inaccurately been associated with a monocytic origin [8,13,14], but recent findings demonstrate that these cells represent bona fide cDC2 populations with unique immune functions [15,19].

***LN cDCs***: Both XCR1^+^ cDC1s and SIRPα^+^ cDC2s are present in most lymphoid and nonlymphoid tissues. We also tested our staining panel on peripheral LN. While LN-resident cDCs reside within these lymphoid organs, cDCs from peripheral tissues travel as migratory cDCs into tissue-draining LNs. Our staining panel allows for the discrimination of these CD11c^hi^MHC-II^+^ LN-resident and CD11c^+^MHC-II^hi^ LN-migratory cDCs (Appendix A).

### 3.2. Identification of Additional Myeloid Cell Populations

In addition to facilitating a comprehensive phenotypic analysis of pDCs and cDCs, the proposed staining panel also enables the identification of other myeloid cell populations in the mouse spleen, including macrophages and monocytes (Table 1). Macrophages present a challenge in distinguishing them from DC populations due to potential contamination and overlapping marker expression. The macrophage marker F4/80 alone may not be sufficient for accurate identification, as certain cDC2 subsets also express this marker [37,38,39,40]. To overcome this issue, CD64, which is highly expressed on macrophages, should be used in combination with F4/80 to allow precise discrimination between macrophages and cDCs (Figure 2B).

It is important to note that the described protocol for isolating DCs will yield only a minor population of macrophages, primarily consisting of F4/80^+^CD64^+^ red pulp macrophages, because splenic macrophages are challenging to isolate. For a more in-depth analysis of splenic macrophage populations, such as marginal zone macrophages (MZMs) and marginal metallophilic macrophages (MMMs), we highly recommend isolating spleen cells using Liberase TL digestion in the presence of lidocaine hydrochloride monohydrate (4 mg/mL) and DNase-1 (50 μg/mL) [41]. MZMs and MMMs can be identified using antibodies against SIGN-R1 and CD169 (Siglec-1), respectively, as well as by their high side scatter and high autofluorescence (Table 3) [41,42].

Monocytes express high levels of CX3CR1 and CD11b (Figure 2I) and can be further divided into Ly6C^−^ and Ly6C^+^ monocytes [43,44] (Figure 2J). Classical inflammatory Ly6C^+^ monocytes selectively migrate to infection sites, where they participate in pathogen clearance, the removal of lipids and dying cells, as well as the production of inflammatory cytokines, and contribute to local and systemic inflammation. In contrast, Ly6C^−^ monocytes patrol the vasculature at steady state to monitor for pathogenic infections [45,46,47]. The molecular mechanisms underlying monocyte subset differentiation, however, remain unclear, and their functional features have not been systematically investigated.

Under (pathogenic) inflammation, monocytes can differentiate into MoDCs, which are heterogenous in phenotype and function. MoDCs are characterized by the expression of CD11c, MHC-II, and monocyte/macrophage-associated markers, including CD64 and Ly6C, while lacking CD26 [15,48]. Consequently, MoDCs can exhibit overlapping phenotypes with certain ESAM^−^ cDC2 subsets (Figure 3), posing challenges for their differential classification, particularly during inflammation [8,13,14,15,19]. Although certain DC subpopulations, likely in mucosal and other peripheral tissues like the skin, may originate from monocytes even during steady-state conditions [8,37,49,50,51], the exact contribution of MoDCs to the splenic DC compartment remains unclear [52]. Using our multicolor FACS panel, no CD11b^+^Ly6C^+^CD64^+^ MoDCs [48] could be detected in the steady-state spleen (Figure 4A). Another DC subset that emerges under certain inflammatory conditions are inflammatory cDC2s (Inf-cDC2s) [15]. This bona fide cDC2 subpopulation can be characterized by the MAR-01 antibody (anti-FcεR1a), identifying them as CD26^+^SIRPα^+^CD64^+^MAR-01^+^ cells within the CD11c^+^MHC-II^+^ DC population. Under steady-state conditions, Inf-cDC2s are absent in the murine spleen (Figure 4B), and further studies are required to determine whether Inf-cDC2s occur outside the lung during inflammation.

## 4. Discussion

The murine spleen harbors heterogeneous DC populations characterized by diverse developmental pathways, phenotypes, and immune-regulatory functions. The flow cytometric analysis of these DCs poses challenges due to the overlapping expression of several surface markers among various myeloid cell populations. Consequently, the accurate identification of individual DC subsets using a limited set of just a handful of markers is almost impossible. This study presents a novel staining panel optimized for the phenotypic characterization of steady-state cDC subpopulations in the mouse spleen, allowing for the unambiguous identification of multiple distinct DC and other myeloid cell populations (Figure 5).

This manuscript highlights the distinct roles and markers of various DC subsets, such as cDC1s and cDC2s. Understanding the unique functions of these subsets can inform the design of targeted vaccines that optimize antigen presentation and stimulate specific immune responses. For instance, in mice Langerin^+^ cDC1 cells are known to be highly effective at cross-presenting antigens to CD8^+^ T cells, making them a strategic target for vaccines aimed at inducing strong cytotoxic T cell responses.

Langerin^+^CD103^+^ cDC1s play pivotal roles in the uptake of apoptotic/dying cells and are therefore essential to priming and shaping CD8^+^ T cell responses [30,31,32]. These cells are involved in the initiation of immunity towards blood-borne pathogens such as viruses, while also contributing significantly to the maintenance of tolerance towards cell-associated self-antigens [29,33,53]. Understanding the ontogeny and differentiation pathways of Langerin^+^ cDC1s, as well as the identification of the human counterpart, is paramount for harnessing their therapeutic potential. However, it is important to note that Langerin expression in these cells occurs primarily intracellularly. Therefore, Langerin might not be a suitable marker for functional experiments due to the requirement of fixing cells with paraformaldehyde. In contrast, CD103, which is co-expressed at high levels with Langerin on cDC1s, could serve as a more feasible marker for functional studies.

Our multiparameter staining panel also underscores the heterogeneity within the cDC2 population, where several subpopulations exhibit an overlapping marker expression with MoDCs and macrophages. This overlap poses challenges, as certain cDC2 subsets, such as ESAM^−^CX3CR1^+^ cDC2Bs, which appear in the mouse spleen upon defective cDC2 homeostasis [13,14,19], as well as inflammatory cDC2s [54], are sometimes misidentified as being of monocytic origin. Currently, there is no optimal strategy for effectively separating these cDC2s from MoDCs [8]. However, the proposed combination of CD26, CD64, Ly6C, and CD64 markers offers a feasible approach to enhance the discrimination and further characterization of these cells, thereby increasing our understanding of their unique immune functions. This advancement is particularly significant given the pivotal role of both cDCs and MoDCs in the development of novel immunotherapies and vaccination strategies, notably in cancer treatment. While the suggested staining panel improves the discrimination between overlapping cDC2 and MoDC phenotypes, its suitability during inflammatory conditions requires further investigation. Determining whether this panel satisfactorily distinguishes these cell types under inflammatory settings is crucial for its broader applicability.

Although we primarily validated this staining panel for steady-state spleen DCs, it is important to note that the staining panel may also be applicable for analyzing the conserved DC network across various tissues, including LN and peripheral tissues such as the lung and intestine. Additionally, while this panel could be potentially used to analyze the DC compartment under inflammatory conditions, it is worth mentioning that these specific conditions might necessitate certain modifications, as we have not yet validated the panel for such scenarios. Furthermore, the staining protocol offers flexibility for customization, for example, to assess the expression of co-stimulatory or inhibitory molecules or to adapt to flow cytometers with fewer channels by selectively replacing or removing certain non-DC core markers.

Comparing murine and human DC populations presents challenges, as many phenotypic markers used to distinguish the different murine DC subsets are not applicable to human DCs [55]. Nevertheless, the human DC network can also be divided into distinct cDC1 and cDC2 subpopulations with unique phenotypic and functional characteristics. In humans, cDC1s are characterized by the expression of CD141 (BDCA3), Clec9a, and XCR1, while human cDC2s are identified by CD1c (BDCA1) expression. Similarly to mice, these cDC2s can be divided into at least two functionally distinct subsets, cDC2As and cDC2Bs [12,56,57]. Notably, human CD141^+^ cDC1s lack Langerin expression, which instead appears at low levels in a CD1a^+^ cDC subset that is closely related to CD1c^+^ cDC2s [58,59,60,61]. Human MoDCs are typically generated from peripheral blood monocytes in vitro. These include both CD14^+^CD16^−^ monocytes, the predominant monocyte population, and CD14^+^CD16^+^ monocytes that closely resemble Ly6^hi^CX3CR1^lo^ classical monocytes in mice. In humans, both monocyte populations can differentiate into MoDCs and acquire DC characteristics, at least during in vitro culture with GM-CSF [62,63]. The differences between mouse and human DC biology underscore the necessity of careful consideration when extrapolating data from animal models to humans. Vaccine development requires human-specific studies to accurately predict vaccine efficacy and immune reactivity.

Finally, for the successful characterization of splenic DCs and other myeloid cell populations, several technical aspects should be taken into careful consideration. Fixation and permeabilization steps, essential for the detection of intracellular antigens, may alter the light scatter properties of cells and increase non-specific background staining. The addition of blocking proteins, such as bovine serum albumin (BSA) or fetal calf serum (FCS), to the staining buffer can reduce this non-specific background. Moreover, the use of Fixable Viability Dyes (FVDs) is recommended to exclude dead cells from the analysis. Notably, certain antigens may be susceptible to epitope masking by paraformaldehyde (PFA) fixation and, therefore, cannot be effectively stained after fixation [64]. While the FACS buffer can be used for preparing the first Antibody Master Mix, we highly recommend using the specialized Brilliant Stain Buffer as described to prevent non-specific reactivity between the polymer-based fluorochromes, which may lead to data under-compensation. Given the calcium-dependent binding of 33D1 (anti-DCIR2), which allows for further cDC2 discrimination, EDTA should be omitted from the FACS buffer when including this antibody in the staining protocol [20,21].

In conclusion, our optimized staining panel, along with its potential future refinements, constitutes a valuable resource for researchers interested in investigating the complex dynamics of the DC network in both health and disease states.

## 5. Conclusions

Given the highly heterogenous nature of DC populations, several important lessons for vaccine development have emerged. First, a comprehensive phenotypic characterization of the different DC populations is crucial to understanding their specific immunomodulatory functions and developmental relationships. This manuscript provides a detailed step-by-step protocol for the multicolor analysis of murine cDC1s, cDC2s, and MoDCs. Second, because inflammation significantly influences DC functions and development, it is essential to consider the impact of the local immune environment when designing vaccines for use in inflammatory or infectious settings, as this can affect DC activation and the overall immune response. Finally, vaccine development must continue to incorporate emerging new findings in DC biology to refine and improve vaccine efficacy.

## Figures and Tables

**Figure 1 vaccines-12-01294-f001:**
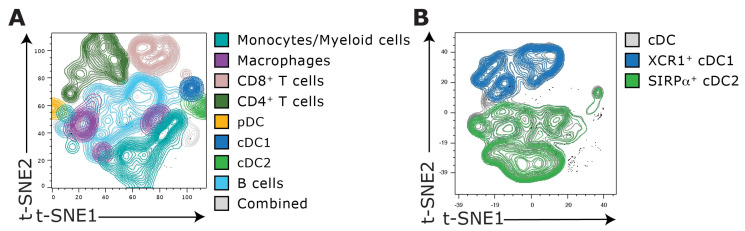
Identification of the major leukocyte and cDC populations in the steady-state murine spleen using a conserved 26-color core-marker FACS panel. (**A**) Unbiased dimensionality reduction analysis via t-stochastic neighborhood embedding (t-SNE) of all living CD45^+^ splenocytes categorizes the major distinct lymphocyte populations (CD4^+^ and CD8^+^ T cells, B cells), myeloid cell populations (monocytes, macrophages), and DC populations (cDC1s, cDC2s, pDCs). (**B**) Mapping of XCR1^+^ cDC1s and SIRPα^+^ cDC2s demonstrates a clear separation of the two main cDC populations in the spleen and illustrates the heterogeneity within each population. Data acquisition was performed using a BD FACSymphony flow cytometer, and data were evaluated using FlowJo software. For t-SNE, CD11c^+^MHC-II^+^ cDCs of three wild-type mice were concatenated in FLowJo with 4.000 events per sample. All markers shown in Table 1, as well as FSC-A and SSC-A, were utilized.

**Figure 3 vaccines-12-01294-f003:**
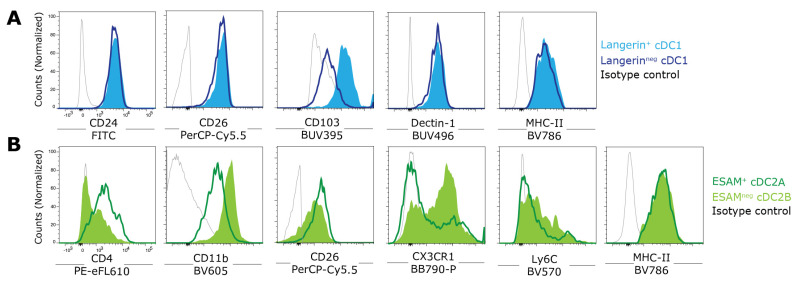
**Phenotypic characterization of cDC1 and cDC2 subpopulations in the murine spleen.** (**A**) Expression of cDC1 markers CD24, CD26, CD103, CD205, and MHC-II on either Langerin^+^ (filled, light blue) or Langerin^−^ (dark blue lines) cDC1s. (**B**) Expression of cDC2 markers CD4, CD11b, CD26, CX3CR1, Ly6C, and MHC-II on either ESAM^+^ (dark green lines) or ESAM^−^ (filled, light green) cDC2s. Single-cell suspensions from the spleen were prepared as described in Section 2.2 and stained as described in Figure 1 and Figure 2. Isotype-matched control antibody stainings are indicated with gray lines in the histogram plots. Data acquisition was performed using a BD FACSymphony flow cytometer, and data analysis was carried out using FlowJo software.

**Figure 4 vaccines-12-01294-f004:**
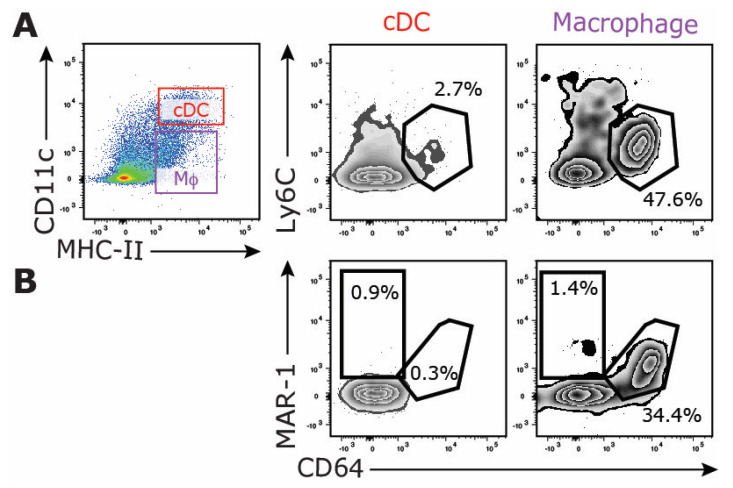
**Phenotypic characterization of MoDCs and Inf-cDC2s in the steady-state murine spleen.** Frequency of Ly6C^+^CD64^+^ MoDCs (**A**) and MAR-01^+^CD64^+^ Inf-cDC2s (**B**) in the murine spleen. DCs were pre-gated as CD11c^hi^MHCII^+^ cells. The gating strategy is based on the expression of either Ly6C^+^CD64^+^ or MAR-01^+^CD64^+^ on macrophages (CD11c^int^MHCII^+^ cells). The FACS plots show one representative mouse/group. Data acquisition was performed using a BD FACSymphony flow cytometer, and data analysis was performed using FlowJo software.

**Figure 5 vaccines-12-01294-f005:**
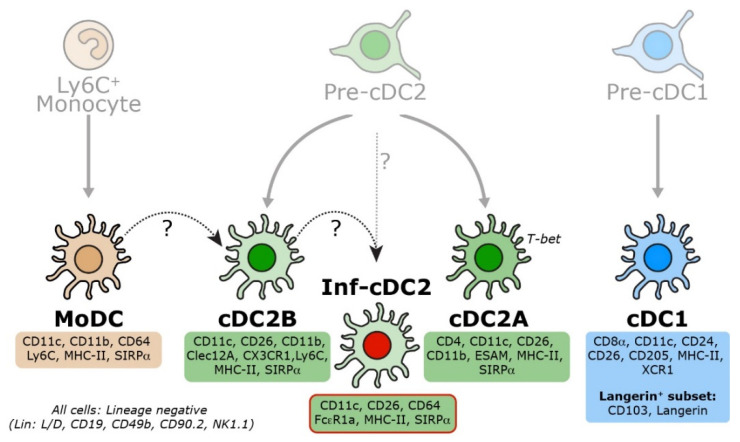
**Characterization of distinct DC populations.** This figure summarizes the differentiation pathways and defining markers of the various DC subsets. Conventional DCs (cDCs) differentiate into two main populations: cDC1s and cDC2s, originating from the common DC precursor-derived pre-DC1s and pre-DC2s, respectively. Monocyte-derived DCs (MoDCs) develop from Ly6C^hi^ monocytes. In lymphoid organs, cDC1s are characterized by the expression of CD8α, CD24, and XCR1. Within the cDC1 population, a subset expressing CD103 and Langerin can be identified. The cDC2 population constitutes at least 2 distinct subpopulations: ESAM^+^ cDC2As and Clec12a^+^ cDC2Bs. The exact relationship between inflammatory cDC2s (Inf-cDC2s) and other DCs remains unclear. Key lineage-defining markers for each DC population are highlighted and identifiable through the outlined protocol.

**Table 1 vaccines-12-01294-t001:** Reagents, antibodies, chemicals, and solutions.

Reagent: Monoclonal Antibodies	Clone	Conjugate	Isotype	Manufacturer	Order Number
Anti-CD4	RM4-5	PE-eFluor 610	Rat IgG2a, κ	Thermo Fisher, Waltham, MA, USA	61-0042-82
Anti-CD8α	53-6.7	PE-Cy7	Rat IgG2a, κ	Biolegend, San Diego, CA, USA	100722
Anti-CD11b	M1/70	BV605	Rat IgG2b, κ	BD Biosciences, Franklin Lakes, NJ, USA	563015
Anti-CD11c	N418	APC-R700	Hamster IgG2	BD Biosciences	565872
Anti-CD19	6D5	PE-Cy5	Rat IgG2a, κ	Biolegend	115510
Anti-CD24	M1/69	FITC	Rat IgG2a	Biolegend	137006
Anti-CD26	H194-112	PerCP-Cy5.5	Rat IgG2a, κ	Thermo Fisher,	45-0261-82
Anti-CD45pan	30-F11	BUV805	Rat IgG2b, κ	BD Biosciences	748370
Anti-CD49b	DX5	PE-Cy5	Rat IgM, κ	Thermo Fisher	15-5971-82
Anti-CD64	X54-5/7.1	PerCP-710	Mouse IgG1, κ	Thermo Fisher	46-0641-82
Anti-CD90.2	30-H12	PE-Cy5	Rat IgG2b, κ	Biolegend	105314
Anti-CD103	M290	BUV395	Rat IgG2a, κ	BD Biosciences	740238
Anti-CD172a (anti-SIRPα)	P84	BUV563	Rat IgG1, κ	BD Biosciences	741349
Anti-CD207 (anti-Langerin)	929F3.01	AF647	Rat IgG2a	Dendritics, Lyon, France	DDX0362A647
Anti-CD317 (anti-PDCA1)	927	BV711	Rat IgG2b, κ	BD Biosciences	747604
Anti-CD369 (anti-Dectin-1)	RA3-6B2	BUV496	Rat IgG2a, κ	BD Biosciences	612950
Anti-CD371 (anti-Clec12A)	1/06-5D3	BV421	Rat IgG2a, κ	BD Biosciences	564795
Anti-CX3CR1	Z8-50	BB790-P	Rat IgG2a, κ	BD Biosciences	Custom
Anti-ESAM	1G8/ESAM	PE	Rat IgG2a, κ	Biolegend	136203
Anti-F4/80	T45-2342	BUV737	Rat IgG2a, κ	BD Biosciences	749283
Anti-FcεR1a	MAR-01	BUV615-P	Hamster IgG	BD Biosciences	751770
Anti-I-A/I-E (anti-MHC-II)	M5/114.15.2	BV786	Rat IgG2b, κ	BD Biosciences	742894
Anti-Ly6C	HK1.4	BV570	Rat IgG2c, κ	Biolegend	128030
Anti-Ly6G	1A8	BV750	Rat IgG2a, κ	BD Biosciences	747072
Anti-MerTK	M1/69	BUV661	Rat IgG2b, κ	BD Biosciences	750679
Anti-NK1.1	PK136	PE-Cy5	Mouse IgG2a, κ	Biolegend	108716
Anti-Siglec-F	E50-2440	BV480	Rat IgG2a, κ	BD Biosciences	746668
Anti-XCR1	ZET	BV650	Mouse IgG2b, κ	Biolegend	148220
Anti-CD16/32 (Anti-FcyRIIB/III)	2.4G2	Purified	Rat IgG2a, λ	Biolegend	101302
**Chemicals, enzymes, and solutions**					
Dulbecco’s Phosphate-Buffered Saline (PBS) without calcium and magnesium				Sigma, St. Louis, MS, USA	D8537
0.5 M Ethylenediaminetetraacetate (EDTA) solution				Sigma	03690
Ethylenediaminetetraacetic acid disodium salt dihydrate				Sigma	E5123-1KG
Sodium hydroxide ≥ 98%, p.a., ISO, in pellets				Carl Roth, Karlsruhe, Germany	6771.1
Brilliant Stain Buffer Plus				BD Biosciences	566385
Fetal Bovine Serum (FBS)				Thermo Fisher, Waltham, MA, USA	10270-106
Fixable Viability Stain (L/D)		FVS780		BD Biosciences	565388
Collagenase Type IV				Worthington, Lakewood, NJ, USA	LS0004186
Deoxyribonuclease I (DNaseI)				Roche, Basel, Switzerland	11284932001

**Table 2 vaccines-12-01294-t002:** Necessary equipment.

Equipment	Company	Purpose
Centrifuge “Z 446 K”	Hermle LaborTechnik, Wehingen, Germany	Centrifugation of 15- and 50 mL tubes, and U-bottom plates
FACSymphony A5	BD Biosciences, San Diego, CA, USA	Flow cytometric analysis of single-cell suspensions
PipetMan (P10-P1000)	Gilson, Middleton, WI, USA	Pipetting
PipetBoy	Thermo Fisher, Waltham, MA, USA	Pipetting
Neubauer chamber 0.100 mm; 0.0025 mm^2^	Superior Marienfeld, Lauda Königshofen, Germany	Cell counting
96-well U-bottom plate (cat# 163320)	Thermo Fisher, Waltham, MA, USA	Sample preparation for flow cytometry
2 mL microcentrifuge tubes (cat# 72.695.200)	Sarstedt, Nümbrecht, Germany	Preparation of Antibody Master Mixes
Pipette tips	Brand, Wertheim, Germany	Pipetting
15 mL tubes (cat# 188271)	Greiner, Kremsmünster, Austria	Centrifugation of cell suspensions
50 mL tubes (cat# 227261)	Greiner bio-one, Kremsmünster, Austria	Centrifugation of cell suspensions
Serological pipettes (1–25 mL)	Gilson, Lewis Center, OH, USA	Pipetting
40 μm cell strainer (cat# 352340)	Falcon, London, UK	Filtration of samples derived before FACS staining
FACS tube (cat#352008)	Sigma, St. Louis, MS, USA	Regular FACS tubes for acquisition of single-cell suspensions derived from spleen at a flow cytometer
Small scissors (cat# 14060-10)	FST, Heidelberg, Germany	Super fine scissors to cut spleen tissue in small parts
Curved forceps (cat# 11271-30)	FST, Heidelberg, Germany	Specialized surgical forceps

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
