# Peer review of "Multiparameter Flow Cytometric Analysis of the Conventional and Monocyte-Derived DC Compartment in the Murine Spleen"

_vaccines, 2024, doi:10.3390/vaccines12111294_

Round 1
Reviewer 1 Report
Comments and Suggestions for Authors
This is a protocol-driven manuscript that also serves as a mini-narrative review of the topic. This is a valuable, outstanding, well-written and -documented manuscript with detailed and useful methods.
The importance of standardizing protocols is critical for the scientific community. This improves our ability to understand changes and perturbations with various interventions, such as vaccines, or with pathology, across difference models. This is of particular importance for dendritic cells and monocytes, whose delineation is difficult to discern even under optimal circumstances, within one lab. Thus, the standardized protocols and rigorous analyses presented within this manuscript are critical additions to the literature and include relatively recent updates (in the last ca. 2 years) to DC, monocyte, macrophage, and MoDc subset markers.
The authors synthesize both pioneering literature on DC subsets as well as the more recently discovered DC and monocyte subset phenotypic markers. This complex topic is thoroughly covered with clarity, and the writing is concise. The flow cytometry data presented are robust and beautifully compensated. One detraction is that the authors' flow cytometry panels are developed for a specific brand and configured flow cytometer, which many labs do not have. However, the authors do provide alternative suggestions and modifications to overcome this obstacle. The dilutions of the antibodies used that is presented in supplementary material is useful despite lot number differences in commercially available antibodies that change concentrations, and thus dilutions used. These dilutions, however, provide dilutions at which titrations can start for different lots of antibodies.
The only suggestions this reviewer has for this already valuable manuscript are the following, below.
1. Perhaps a paragraph on the differences in human markers from those outlined for mouse model could be added to the discussion section, for those researchers pursuing clinical studies and wish to adapt and develop the panels outlined, for human studies. A few sentences on the points of divergence between the protocol presented and adaptations that may be important for human studies could be added.
2. A summary graphic of a flow chart that would graphically represent the various DC, monocyte, macrophage, and Mo-DC populations delineated by the protocol, displaying their associated markers graphically. While a methods manuscript does not necessarily include this material, this could improve reader comprehension of the myeloid subsets identified by the outlined protocol.
Overall, this is a remarkable methods paper that has elements of a mini-review and is of significant interest to the scientific community pursuing myeloid cell studies for vaccines, cancer therapies, and infectious disease.
Minor Comments:
Line 14 “very” does not add much to this statement
Line 29 take “their” out
Line 29 “functions” take out “functional properties”
Line 50 “To this end, here we present…”
Line 54 This could be a listed as a subheading such as Rationale
Line 126 Add to “until use” – add a specific time frame when they can be analyzed, i.e., in next 48 hours?
Line 152 Perhaps say why Thy 1.2 is used here (earlier than later in text, mentioned in 352) instead of CD3 (for all mouse strains?)
Line 186 Change to “analogous to T cells”
Line 352 This seems unusual – PFA epitope masking for standard markers like CD3 or CD4?
References Perhaps Minutti et al., Nat Immunology Feb 2024 can be added
Author Response
Please see enclosed point-by-point reply to reviewer 1.

Reviewer 2 Report
Comments and Suggestions for Authors
Peer review for manuscript ID vaccines-3250774: Backer et al. “High-parameter flow cytometric analysis of the conventional and monocyte-derived DC compartment in the murine spleen”. This manuscript is being reviewed for possible publication in Vaccines journal.
This paper is a review of the current designs, development and implementation of a multiparameter, multicolor flow cytometric procedure to identify DC subpopulations from murine spleen samples. The authors appear to understand the subject matter and as a flow cytometry expert, I can state that they provide a very compelling method to tease out the various cellular populations of DC. However, their paper does not provide any mention of vaccines synthesis or usage based on this protocol. There are two small tie-ins to vaccines in general in Lines 41 and 329. These sentences alone do not render the manuscript as a vaccines paper. The authors would be better to submit their paper to Cytometry or Biotechniques journals or perform a video and submit to JoVE, provided they adhere to my critiques below. Unfortunately, I am rejecting this manuscript for Vaccines journal.
Grammar Suggestions
1. Line 87: Change “0,5” to “0.5”.
2. Line 126: Delete “in the fridge” and replace with “at 4ºC”. / Insert “ready to” before “use”.
3. Line 128: Change “at” to “with”.
4. Lines 105, 117: Do not hyphenate “upside down”.
5. Line 150: Make “Figures” singular.
6. Figure 2: Replace “,” with decimal point in values given in the panels (refer to comment for Line 87 above).
7. Line 170: Do not capitalize “viability”.
8. Line 243: Capitalize “figures”.
Supplemental document: There are a couple of grammar suggestions in this document. See uploaded document.
Science Suggestions
Title
1. Line 1: Change “High-parameter” to “Multiparameter”. While high-dimensionality analysis is used for the authors’ data, the term “high” in flow cytometry is more indicative of the concentration frequency of labeled cell types in a sample(s).
Introduction
2. Lines 41-45 and 54-58: The authors understand that due to the “phenotypic overlap” of DC subpopulations, it is very difficult to differentiate and identify them using molecular biological techniques. On the other hand, the authors do not talk about single cell sorting analysis of their identified DCs that would lend credence to the data shown. Also, they briefly mention single cell suspensions in the Fig. 3 legend. If the authors employed a flow cytometric single cell sorting strategy, then they should talk about this in the M&M, Results, and Discussion sections.
Materials and Methods
3. Lines 96-112: Authors need to put in an introductory sentence that they are providing the information for the staining kit and its reagents in the Supplemental document. Furthermore, they should list this information in the tables they generated for the supplies. / Authors need to include citation(s) for all protocol subsections in M&M.
4. Lines 64-70, Tables 1 and 2: While the authors have been studious about providing ordering information for the reagents utilized in their experiments, they should also consider providing the location of the companies from which they ordered (i.e. city, country) for completeness.
5. Line 71, 72: Where were the C57BL/6 mice obtained? While the age of the mice is given in Line 168, this information should be included here as well.
6. Lines 76-93: Citations are required for the isolation procedure used.
7. Lines 96-112: Please list the name of the kit you are using with company information provided as well.
8. Line 105: Company and ordering information needed for the blocking agent. / Also, what volume/concentration was used?
9. Lines 125-128: Authors need to explain the cytometer set up. This is crucial information since there is variability across machines as well as sample preparation. Authors must answer the following questions: (a) What optical filters are being used for each color/stain? (b) What is the flow cytometer data acquisition targeted on: FSC vs SSC (Fig. 1) for example? (c) Was the data collected, color compensated before or after acquisition or both? These are crucial questions that should be addressed when investigating rare events (subpopulations). This information provides the reader with sufficient background information to understand more fully the data being presented. / Flow cytometer and FloJo company information needs to be provided.
10. Lines 127: Do the authors mean 35 mM filters? If so, then ordering and company information should be provided.
Results
11. Line 146: The term “4.000 cDC” needs to be clarified.
12. Figure 2: Authors should include log dimensions on axes as shown for the Live/Dead FVS780 histogram. This aids the reader in understanding the clusters position in 2D/3D space and provides a visual reference for concentrations of the many subpopulations.
13. Lines 242, Figure 3: Exactly how were single cell suspensions obtained? Please see my comments under the Introduction section above.
Discussion
14. Line 319: Change “high-parameter” to “multiparameter”.
15. Lines 324, 325: Citation is needed for this statement.
16. Lines 338-341: Are the authors going to validate their panel for inflammatory conditions? It would be nice to see if they are working on this.
17. Lines 351-353: Have the authors considered a non-PFA fixative such as Cyto-Chex® BCT? This fixative can be utilized effectively in the preservation/expression of cellular markers where PFA be is inappropriate.

English was very good - just a few minor suggestions.
Author Response
Please see enclosed point-by-point reply to reviewer 2.

Reviewer 3 Report
Comments and Suggestions for Authors
This contribution represents an understandable, excellent written and presented protocol for the flow-cytometric characterization of different dendritic cell (DCs) subpopulations in mice.
It is accompanied by pertinent, very informative and well designed figures and tables, as well by a good number of references.
Maybe, a detailed reference/characteristics of the Fc-block solution should be given.
Author Response
Please see enclosed point-by-point reply to reviewer 3.

Round 2
Reviewer 2 Report
Comments and Suggestions for Authors
This is an excellent paper and very well written. I highly recommend publication.